# Ensemble-based Deep Reinforcement Learning for Vehicle Routing Problems under Distribution Shift

**Yuan Jiang**
School of Computer Science and Engineering
Nanyang Technological University
yuan005@e.ntu.edu.sg

**Zhiguang Cao**
School of Computing and Information Systems
Singapore Management University
zgcao@smu.edu.sg

**Yaoxin Wu**
Department of Industrial Engineering & Innovation Sciences
Eindhoven University of Technology
y.wu2@tue.nl

**Wen Song**[*]
Marine Science and Technology
Shandong University
wensong@email.sdu.edu.cn

**Jie Zhang**
School of Computer Science and Engineering
Nanyang Technological University
zhangj@ntu.edu.sg

## Abstract

While performing favourably on the independent and identically distributed (i.i.d.) instances, most of the existing neural methods for vehicle routing problems (VRPs) struggle to generalize in the presence of a distribution shift. To tackle this issue, we propose an ensemble-based deep reinforcement learning method for VRPs, which learns a group of diverse sub-policies to cope with various instance distributions. In particular, to prevent convergence of the parameters to the same one, we enforce diversity across sub-policies by leveraging Bootstrap with random initialization. Moreover, we also explicitly pursue inequality between sub-policies by exploiting regularization terms during training to further enhance diversity. Experimental results show that our method is able to outperform the state-of-the-art neural baselines on randomly generated instances of various distributions, and also generalizes favourably on the benchmark instances from TSPLib and CVRPLib, which confirmed the effectiveness of the whole method and the respective designs.

## 1 Introduction

The vehicle routing problem (VRP) is a type of combinatorial optimization problems (COPs) with significant real-world applications, especially in the field of logistics [1]. The goal is to optimally route a fleet of vehicles, originating at a depot, to serve a set of clients. Given the NP-hard nature, heuristic methods are usually preferred to solve VRPs in practice. However, traditional heuristics always treat each VRP instance independently, resulting in limited computational efficiency or solution quality [2]. Hence, there is growing interest in developing neural heuristics based on deep (reinforcement) learning to exploit the underlying patterns across VRP instances and then generalize to unseen ones [3].

Early neural heuristics for VRPs were primarily supervised [4, 5], whose performance depends heavily on the ground-truth labels, making them less desirable due to the expensive computation for

---

[*]Corresponding Author

37th Conference on Neural Information Processing Systems (NeurIPS 2023).

obtaining optimal solutions. By contrast, deep reinforcement learning (DRL) relies on feedback from the environment rather than the ground-truth optimal solution as a reward, allowing neural heuristics to perform favorably in solving VRPs [6]. Typically, DRL can be used to train networks for selecting the next client (or node) to visit in VRPs [7, 8, 9, 10], which is more capable of generalizing to different problem sizes in comparison with the supervised counterparts [11, 12].

While DRL-based neural heuristics have achieved a series of success for VRPs, most of them perform reasonably well only when the underlying deep models are trained and tested on the independent and identically distributed (i.i.d.) instances with respect to the client locations, e.g., the ones generated using the uniform distribution. In the presence of distribution shifts, they struggle to deliver desirable performance, which greatly hinders the applications of neural heuristics into practice. Especially in real-world VRP instances, the client locations may vary dramatically according to weekday or weekend, sunny or rainy weather, etc, leading to various or even unknown distributions. Therefore, how to strengthen the cross-distribution generalization capability is of great importance to the neural heuristics for VRPs. Although a number of preliminary studies have been conducted to investigate this issue, including the distributionally robust optimization (DRO, [13]), curriculum-learning-based hardness-adaptive generator (HAC, [14]), and adaptive multi-distribution knowledge distillation (AMDKD, [15]), they are still far from optimal due to their inferior performance or less practical settings. Hence, it remains challenging to comprehensively and effectively expand the limited generalization capability of the neural heuristics against the distribution shifts for VRPs.

Motivated by the recognition of the limited capability of an inflexible model to learn from multiple distributions [16], and inspired by the fact that an ensemble of varied models can leverage their respective strengths to collaboratively handle different distributions [17, 18], we propose an end-to-end ensemble-based deep reinforcement learning (EL-DRL) to solve VRPs in the presence of distribution shifts, which ensures diverse and complementary learning across different sub-models (or sub-policies), thereby fostering the cross-distribution generalization capability. Specifically, we first extend the REINFORCE algorithm [19] and policy gradient [20] in DRL to the ensemble setting by adaptively training a set of sub-policies on instances of mixed distributions to collaboratively solve a VRP instance. Then, to encourage diversity among the sub-policies, we harness Bootstrap with random initialization to enforce that each sub-policy is updated according to different loss signals. Additionally, we also exploit different regularization terms to prevent the collapse of sub-policies by explicitly pursuing inequality among them.

Accordingly, our contributions are outlined as follows. (i) We empirically demonstrate that a simple ensemble of similar sub-policies leads to limited performance in solving VRPs with distribution shifts. (ii) We propose an ensemble-based deep reinforcement learning (EL-DRL), and its ensemble-based policy gradient allows multiple sub-policies to learn to collaboratively leverage their strengths on respective distributions. We foster diversity by assigning different loss signals to sub-policies and applying regularization based on inequality metrics. (iii) Our EL-DRL adaptively trains sub-policies on instances of various distributions and is flexible without manually specifying instance class or group [13, 15] to train each sub-policy. (iv) We deploy POMO [9] as the sub-model in EL-DRL for solving the traveling salesman problem (TSP) and capacitated vehicle routing problem (CVRP) on both synthetic and benchmark instances of various distributions. The results verified the superior cross-distribution generalization of our EL-DRL.

## 2 Related Works

### 2.1 Deep Learning for Vehicle Routing Problems

In recent years, there has been extensive exploration of the use of deep (reinforcement) learning to solve vehicle routing problems. The Pointer Network (PtrNet) [4], originated from the encoder-decoder-structured sequence-to-sequence network in natural language processing (NLP), was employed to solve VRPs using supervised learning. To curtail the computational expense of obtaining ground-truth labels for VRPs, a follow-up study [21] trains the deep architecture using reinforcement learning. Performance was further improved by introducing a self-attention encoder [22] and an attention decoder, leading to the prominent Attention Model (AM). The POMO [9] was constructed upon the Attention Model, which augmented input instances and started inference from multiple cities. Three distinct strategies were presented for integrating active search into POMO in the inference phase [23], resulting in significantly improved performance. Graph neural networks (GNNs) and their

variants, such as graph convolutional networks (GCNs) and graph attention networks (GATs), have been applied to the graphs of VRP instances [24, 25, 26, 27, 5]. A concurrent line of improvement methods [28, 29, 30, 31, 10] focuses on refining an initial solution through iterative local rewriting. Neural methods based on deep learning have shown impressive results when training and testing instances share the same or similar distribution with respect to node locations, such as a uniform distribution [9, 32]. However, applying the pre-trained models of these methods to instances with non-uniform or complex distributions can result in significant gaps to optimal solutions [33, 14]. To enable the generalization beyond a single uniform distribution during training, group distributionally robust optimization (DRO, [13]) has been used to train neural models such as POMO and GCN on different distributions, which requires manual labeling of typical and atypical instances and specifying distribution groups. A curriculum-learning-based AM (HAC, [14]) has also been explored, which is trained on instances of varying hardness produced by an adaptive generator but limited to the mixed-Gaussian distribution. More recently, a study (AMDKD, [15]) distills the knowledge from various pre-trained teacher models, which manually assigns specific distribution to train each teacher model. Multi-view graph contrastive learning (MVGCL, [34]) is also applied to incorporate generalizable representations in policy learning for better knowledge transferability across distributions.

## 2.2 Deep Ensemble Learning

Deep ensemble learning models combine the strengths of both deep learning and ensemble learning to achieve superior generalization performance [35, 36]. Ensemble learning is recognized as a promising solution for adaptation to various tasks [37, 35, 36], due to the unique advantage that respective sub-models hold in solving different tasks. A few works have (implicitly) borrowed ideas from ensemble learning to solve VRPs [38, 29]. For example, MDAM [38] leverages the AM equipped with multiple decoders to solve VRPs, and each decoder corresponds to a sub-model in ensemble learning. However, due to the demanding computation, no mechanism is in place to ensure the diversity of constructed solutions beyond the first node. L2I [29] requires manually assigning modification rules to train different policies separately, which also takes prohibitively long runtime to search for better solutions iteratively. In summary, these works focus more on combining multiple models rather than promoting the automatic acquisition of diverse knowledge. On the other hand, diversity is crucial in ensemble learning for improving prediction performance and robustness [39, 40]. Ensemble diversity measures, algorithms for creating diverse ensembles, and ensemble consensus have been investigated to produce high-accuracy ensemble outputs [41, 42]. Our work differs from existing DRL-based methods for VRPs in that we explicitly consider and enforce diversity among sub-models, allowing them to capture different patterns from various distributions without much manual interference.

## 3 Methodology

In this section, we present how our ensemble learning paradigm trains a set of diverse DRL-based sub-policies, i.e., EL-DRL (see Figure 1), to solve VRP instances of various distributions effectively.

### 3.1 Preliminary

The traveling salesman problem (TSP) and the capacitated vehicle routing problem (CVRP) are the two most classic VRP variants. The TSP involves finding the optimal route to visit a set of $N$ cities (or nodes) $\{v_1, v_2, \ldots, v_N\}$, with the objective to minimize the total distance traveled. The CVRP considers a fleet of (homogeneous) vehicles with limited capacity, where the vehicle must only visit a client once and also satisfy its demand. The objective is to minimize the total distance traveled by all vehicles while complying with the capacity constraint.

Our EL-DRL originates from the conjecture that high similarity in parameters of the neural network in the ensemble-based learning paradigm correlates to poor performance, as demonstrated in Section 4.4. Recent studies on ensembles with reinforcement learning [43, 44, 45] have primarily focused on utilizing deep Q-learning [46] to play games with a limited number of actions determined by the game controller. However, estimating Q-values for VRPs with hundreds of potential actions (i.e., the next node to visit) is challenging [26, 47], and the performance of these methods is often inferior to that of policy gradient [48]. We bridge this gap by designing the ensemble-based DRL (EL-DRL) to solve VRPs with policy gradient. While we build EL-DRL on top of the policy optimization with multiple optima (POMO) [9], our method is generic and can also be applied to other neural heuristics.

For clarity, we begin by explaining the process of using a single policy to solve a TSP instance through reinforcement learning. In specific, the policy gradient aims to learn a policy $\pi_\theta$ represented by a neural network and parameterized by $\theta$, which constructs a solution for the input instance $s$. The (partial) solution $\tau_t$ is defined by the sequence of selected actions $a_1, a_2, \ldots, a_N$. Typically, the neural network employs an encoder to embed the input features (e.g. coordinates of nodes) and a decoder to sample the next action $a$ [8, 10]. To solve TSP, the decoder autoregressively outputs the probability distribution $\pi_\theta (\cdot \mid s, \tau_t)$ for selecting a valid node (action) to visit at each step $t \in \{1, \ldots, N\}$ such that $\pi_\theta(\tau) = \prod_t \pi_\theta (a_{t+1} \mid s, \tau_t)$. Once a complete solution $\tau$ has been constructed, the negative length of the trajectory can be computed and used as the reward $R(\tau)$.

## 3.2 Ensemble-based Policy Gradient

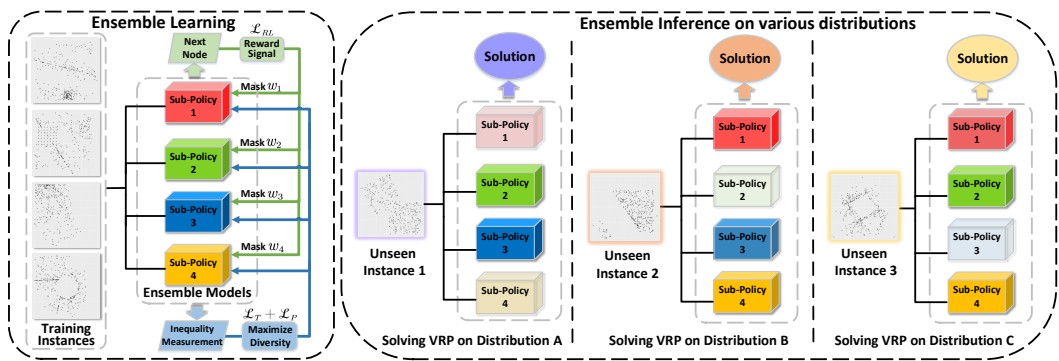

Figure 1: Illustration of Our EL-DRL. It trains sub-policies using respective masked reward (reinforcement loss) signals and inequality regularization to maximize the diversity. During inference, sub-policies synergize by leveraging their strengths on various distributions. Different levels of transparency indicate varied contributions of each sub-policy in deriving the solutions.

Our architecture employs a shared attention encoder from AM [8], followed by $M$ respective attention decoders, as illustrated in Figure 1. Each sub-policy, denoted by $\pi_\theta^m$ with parameters $\theta_m$, corresponds to a unique decoder plus an encoder that shares parameters with other sub-policies. In this way, the overall representation of an instance is only calculated once through the (computationally intensive) encoder. The subsequent lightweight decoders reuse this representation and collaboratively select nodes autoregressively, reducing the computational overhead for the ensemble training.

In our EL-DRL, the action $a_t$ at each step is determined by gathering output from all sub-policies, i.e, $\pi_\theta(a_{t+1} \mid s, \tau_t) = \frac{1}{M} \sum_1^M \pi_\theta^m(a_{t+1} \mid s, \tau_t)$. This is inspired by the real-world practice where open-source communities combine the strengths of different individuals to achieve various common goals [49]. Similarly, our ensemble policies leverage the strengths of individual sub-models, e.g., some of them may perform well on Uniform distribution, while others excel on Gaussian distribution. Consequently, EL-DRL is not dominated by any individual sub-policy. Instead, it averages the probabilities produced by all sub-policies to determine an action at each step.

Using complex RL algorithms may not guarantee significant improvement. A better practice is to tailor learning algorithms for routing problems. We train the sub-policy with the REIN-FORCE algorithm [19] based on the POMO model[9], which samples a set of solution trajectories $\{\tau^1, \tau^2, \ldots, \tau^N\}$ that start from each of all nodes ($i \in \{1, 2, \ldots, N\}$), and gather each return $R(\tau^i)$ (i.e., the negative of route length). The goal is to maximize the reinforcement learning objective $\mathcal{L} = \mathbb{E}_{\tau^1 \sim \pi_\theta} R(\tau^1) \mathbb{E}_{\tau^2 \sim \pi_\theta} R(\tau^2) \ldots \mathbb{E}_{\tau^N \sim \pi_\theta} R(\tau^N)$ by updating $\theta$ to increase the likelihood of sampling desirable trajectories in the future. We extend this objective to multiple sub-policies, which maximizes the expected return $\mathcal{L}_{RL}$ with gradients computed as below,

$$\nabla \mathcal{L}(\theta \mid s) \approx \frac{1}{NM} \sum_{i=1}^N \sum_{m=1}^M \left( R\left(\tau_m^i\right) - b(s) \right) \nabla \log \pi_\theta^m \left(\tau_m^i \mid s\right), \tag{1}$$

where $b(s)$ is a baseline to reduce the gradient variance, $\pi_\theta^m \left(\tau_m^i \mid s\right)$ means the probability produced by the $m$-th sub-policy for the solution $\tau_m^i$ given the instance $s$. Our sub-policies synergize to generate a single trajectory for each starting node $i$, thus we have $\tau_m^i = \tau^i$ and $R(\tau_m^i) = R(\tau^i)$ for all $m$.

While it is feasible to train all the sub-policies together with Eq. (1), directly applying it to them may lead to the identical convergence of parameters, because all sub-policies receive the same reward which is used to compute the reinforcement loss signal. To alleviate this issue, we harness the Bootstrap with random initialization [50] for training the ensemble of sub-policies, which fosters diversity across the sub-policies via two mechanisms. Firstly, we initialize all decoders with distinct random parameters, inducing an initial diversity in the ensemble. Secondly, we leverage distinct reinforcement loss signals to train each sub-policy. In particular, for a sub-policy $\pi_\theta^m$, we generate binary masks $w_m^i$ from a Bernoulli distribution with parameter $\beta \in (0, 1]$ and then apply them to the rewards. Integrating those designs into Eq. (1), we arrive at:

$$\nabla \mathcal{L}_{RL}(\theta \mid s) \approx \frac{1}{NM} \sum_{i=1}^{N} \sum_{m=1}^{M} w_m^i \left( R\left(\tau^i\right) - b(s) \right) \nabla \log \pi_\theta^m \left(\tau^i \mid s\right). \tag{2}$$

When updating the decoder parameters of sub-policies, we also apply the bootstrap mask to each objective function, similar to the above one in Eq. (2). Additionally, we utilize a shared baseline $b(s)$ averaged from all sampled trajectories:

$$b(s) = \frac{1}{NM} \sum_{i=1}^{N} \sum_{m=1}^{M} R\left(\tau_m^i\right) = \frac{1}{N} \sum_{i=1}^{N} R\left(\tau^i\right). \tag{3}$$

The probabilities of producing the actions of $\tau^i$ are reinforced through other $N$-1 unique trajectories. This shared baseline contributes to reducing gradient variance, and also potentially helps restrain the identical convergence with respect to the sub-policies [9].

### 3.3 Diversity Enhancement via Regularization

To further promote the diversity of the ensemble of sub-policies, we exploit the regularization to encourage explicit diversity across the parameters $\theta^1, \ldots, \theta^M$ of sub-policies during training. In order to prevent sub-policies from converging to the same parameters or simply duplicating each other, we need to push them away from each other in the parameter space.

This motivates us to measure and pursue the inequality between sub-policies. Therefore, we utilize a statistical measurement of inequality within a population (ensembles) in economics, i.e., the Theil index [51]. From the perspective of information theory, it is also a measurement of redundancy, which indicates the difference between the maximum possible entropy of the data and the observed entropy [42]. With the Theil index, we could measure whether our ensembles have learned their respective knowledge instead of imitating each other. Concretely, the first regularization term based on the Theil index in our loss function is expressed as $\mathcal{L}_T = \frac{1}{M} \sum_{m=1}^{M} \frac{x_m}{\mu} \ln \frac{x_m}{\mu}$, where $x_m$ is the $\ell^2$-norm of $\theta_m$ and $\mu$ is the mean of all $\ell^2$-norms.

Furthermore, since we rely on an ensemble of sub-policies to make the decision, we could also explicitly exert a penalty that discourages all the individual sub-policies from converging toward a common value. This inspires us to utilize the second regularization term from the principles of consensus optimization [52], where multiple models are independently optimized with their own task-specific parameters. Formally, this regularization term is defined as $\mathcal{L}_P = \sum_{m=1}^{M} \|\bar{\theta} - \theta^m\|^2$, where $\bar{\theta}$ is the mean of the parameters for all sub-policies.

Notice that we intend to minimize the objective function for training our EL-DRL, thereby we integrate the regularization terms with negative signs to maximize the ensemble diversity (inequality). The whole loss function, inclusive of regularization terms with their coefficients ($\alpha_1$ and $\alpha_2$) for our EL-DRL is expressed as:

$$\nabla \mathcal{L} \approx \mathcal{L}_{RL}(\theta \mid s) - \alpha_1 \mathcal{L}_T - \alpha_2 \mathcal{L}_P. \tag{4}$$

At last, the parameters of policies are updated using gradient ascent based on Eq. (4). In the inference phase, EL-DRL collaboratively and greedily produces multiple trajectories by augmenting each input instance with rotations [9] and starting the trajectory from each of all nodes. The final solution is specified as the best one among all the sampled trajectories (solutions).

# 4 Experiments

We propose EL-DRL to train multiple diverse policies for enhancing cross-distribution generalization performance. To evaluate the robustness of our method in handling distribution shifts, EL-DRL and other baselines are tested on synthetic TSP and CVRP instances from various distributions, together with benchmark instances from TSPLib [53] and CVRPLib [54]. Additionally, we perform ablation experiments to demonstrate the importance of diversity and the key designs of our EL-DRL.

**Baselines.** For TSP experiments, we evaluate against the Concorde exact solver [55] and representative end-to-end neural approaches: Attention Model (AM) [8], Policy Optimization with Multiple Optima (POMO) [9], Multi-Decoder Attention Model (MDAM)[38], Distributionally Robust Optimization with POMO (DROP) [13], Hardness-Adaptive Curriculum (HAC) [14], and Adaptive Multi-Distribution Knowledge Distillation (AMDKD) [15]. Of these methods, DROP, HAC and AMDKD are recent approaches tackling the cross-distribution generalization issue for solving VRPs. As HAC was initially intended only for TSP, we did not take it into account for our CVRP experiments.

**Implementation.** EL-DRL employs an architecture similar to AM[8] but with one encoder and four decoders as the backbone. Preliminary experiments showed that the use of four decoders allows for a good balance between computational overhead and performance. Each sub-policy was trained using the same hyperparameters as the original POMO [9]. We use Adam optimizer with a learning rate of 1e-4. The values of $\alpha_1$ and $\alpha_2$ are tested and set according to testing sets. All the experiments of neural baselines are conducted in parallel on NVIDIA GTX 2080Ti GPUs for both training and inference. The training time for EL-DRL varied with the problem size. Taking CVRP100 as an example, one epoch takes about 5 minutes for EL-DRL. We applied early stopping during training when the reduction in the gap was not significant.

The optimal solution for TSP is calculated using the Concorde solver.[2] For CVRP, as existing solvers are difficult to obtain optimal solutions within an acceptable time frame, we employ the strong heuristic HGS[3] [56] as the traditional baseline instead, which has reported better performance [57] than LKH3 [58]. Regarding neural baselines, we reuse their open-sourced code from GitHub. For AM-based baselines, we sample 1,280 solutions following [8]; for MDAM we set beam search size to 50; for POMO-based baselines, we adopt the greedy rollout with augments following [9], which produces multiple greedy trajectories by rotating each input instance to generate $8\times$ augmented instances with $8$ different angles but the same optimal solution. For each generated instance, it starts the trajectory from each of the $N$ nodes. In this way, POMO-based baselines sample $8 \times N$ solutions in total for each input instance.

**Dataset.** To ensure the hardness and variety of training distributions and to guarantee that the training instances are not seen during testing, we generate 7.2 million training instances from mixed distributions. Data generation is done by first sampling a uniform instance and then non-repetitively applying three randomly mutation operators[4] of TSPGEN to the same instance. So a single training instance is of mixed distributions and does not belong to any single distributions in the testing. During the inference, unlike most existing neural heuristics that are only tested on a uniform distribution, we evaluated all methods on various distributions, including Explosion, Compression, Cluster, Expansion, and Rotation, to assess the generalization performance against distribution shift. These distributions are more diverse in terms of both visual and quantitative measures compared to the uniform distribution [59], making it fairly challenging for neural heuristics to generalize. We test baselines on 10,000 instances in total, with each of the five testing distributions containing 2,000 instances. Therefore, the training instances do not fall into any type of testing distribution and all the test instances are sampled from distribution not seen during training. HAC is trained on instances created by the built-in data generator as its design relies on this process. The training of DROP requires class labels of the input instance and we specify the class of the last applied mutation operation for it. AMDKD requires manually assigning a single distribution for training each teacher model, which is less applicable to our setting where each single training instance is of mixed distributions. Thus, we follow its original setting for the best performance, and directly utilize the pre-trained AMDKD to solve instances from TSPLib and CVRPLib (Table 5 ).

---

[2]https://www.math.uwaterloo.ca/tsp/concorde.html
[3]https://github.com/chkwon/PyHygese
[4]https://github.com/jakobbossek/tspgen/tree/master/R

## 4.1 Cross-distribution Generalization on TSP

Table 1: Results of cross-distribution generalization on TSP50

| Distribution | Metric | Concorde | AM | POMO | MDAM | HAC | DROP | EL-DRL |
|---|---|---|---|---|---|---|---|---|
| **Explosion** | Len. | 4.74 | 4.88 | 4.84 | 4.87 | 4.85 | 4.88 | **4.83** |
| | Gap | 0.00% | 2.95% | 2.11% | 2.74% | 2.32% | 2.95% | **1.90%** |
| **Compression** | Len. | 5.22 | 5.37 | 5.33 | 5.36 | 5.35 | 5.34 | **5.32** |
| | Gap | 0.00% | 2.87% | 2.11% | 2.68% | 2.49% | 2.30% | **1.92%** |
| **Cluster** | Len. | 5.37 | 5.56 | 5.50 | 5.54 | 5.53 | 5.52 | **5.48** |
| | Gap | 0.00% | 3.54% | 2.42% | 3.17% | 2.98% | 2.79% | **2.05%** |
| **Expansion** | Len. | 4.44 | 4.58 | 4.55 | 4.57 | 4.58 | 4.60 | **4.53** |
| | Gap | 0.00% | 3.15% | 2.48% | 2.93% | 3.15% | 3.60% | **2.03%** |
| **Rotation** | Len. | 4.54 | 4.69 | 4.64 | 4.67 | 4.66 | 4.64 | **4.63** |
| | Gap | 0.00% | 3.30% | 2.20% | 2.86% | 2.64% | 2.20% | **1.98%** |
| **Avg. Inf. Time (s)** | | 0.08 | 0.07 | 0.01 | 0.04 | 0.08 | 0.01 | 0.02 |

Table 2: Results of cross-distribution generalization on TSP100

| Distribution | Metric | Concorde | AM | POMO | MDAM | HAC | DROP | EL-DRL |
|---|---|---|---|---|---|---|---|---|
| **Explosion** | Len. | 6.09 | 6.31 | 6.22 | 6.29 | 6.38 | 6.27 | **6.21** |
| | Gap | 0.00% | 3.61% | 2.13% | 3.28% | 4.76% | 2.96% | **1.97%** |
| **Compression** | Len. | 6.89 | 7.16 | 7.06 | 7.12 | 7.18 | 7.12 | **7.03** |
| | Gap | 0.00% | 3.92% | 2.47% | 3.34% | 4.21% | 3.34% | **2.03%** |
| **Cluster** | Len. | 7.26 | 7.58 | 7.45 | 7.53 | 7.63 | 7.46 | **7.41** |
| | Gap | 0.00% | 4.41% | 2.62% | 3.72% | 5.10% | 2.75% | **2.07%** |
| **Expansion** | Len. | 5.57 | 5.80 | 5.72 | 5.78 | 5.88 | 5.74 | **5.69** |
| | Gap | 0.00% | 4.13% | 2.69% | 3.77% | 5.57% | 3.05% | **2.15%** |
| **Rotation** | Len. | 6.02 | 6.28 | 6.17 | 6.24 | 6.34 | 6.23 | **6.14** |
| | Gap | 0.00% | 4.32% | 2.49% | 3.65% | 5.32% | 3.49% | **1.99%** |
| **Avg. Inf. Time (s)** | | 0.50 | 0.22 | 0.02 | 0.16 | 0.23 | 0.03 | 0.04 |

We evaluated the cross-distribution generalization of our EL-DRL on TSP50 and TSP100. Tables 1 and 2 gathered the averaged tour length and gaps of all baselines compared to the exact solver Concorde on unseen instances from five different distributions. Although all baselines were trained on mixed distributions, they did not generalize well to other specific distributions (especially for AM), resulting in drastically deteriorated performance (about 2.5% to 4% gaps) compared to the i.i.d testing results they reported in the original paper. For example, POMO reports only 0.03% and 0.14% gaps compared to LKH3 [58] on TSP50 and TSP100, respectively, when trained and tested on a uniform distribution in [9]. However, its gaps increase up to 2.69% in our cross-distribution experiments. This phenomenon implies that these neural baselines are relatively incapable of learning from the underlying common patterns on complex training distributions. In contrast, our EL-DRL significantly improved generalization performance on the five respective testing distributions, achieving the smallest gap among neural heuristic baselines. For example, our EL-DRL reduced the gaps by 2.34% (2.07% vs. 4.32%) compared to AM and achieved a 0.55% (2.07% vs 2.62%) gap reduction on the TSP100 with Cluster distribution compared to the best neural baseline POMO. Notice that while multiple decoders in MDAM slightly improved its performance compared to AM, it failed to generalize to various distributions with gaps exceeding 3.2% on TSP100. This indicates that simply composing multiple models (even with KL divergence) does not necessarily result in good generalization performance. Meanwhile, EL-DRL also consistently beats the recent generalization-specialized DROP and HAC in terms of both gaps and time efficiency. Especially, DROP exhibits unstable generalization performance and HAC does not generalize well on TSP100 (4.21-5.32% gaps), possibly due to special data processing requirements for training, which hinder their application to our scenario. In short, our EL-DRL outperforms these two state-of-the-art approaches in generalizing to cross-distribution.

Table 3: Results of cross-distribution generalization on CVRP50

| Distribution | Metric | HGS | AM | POMO | MDAM | DROP | EL-DRL |
|---|---|---|---|---|---|---|---|
| Explosion | Len. | 9.79 | 10.02 | 9.92 | 9.98 | 9.91 | **9.88** |
|  | Gap | 0.00% | 2.35% | 1.33% | 1.94% | 1.23% | **0.82%** |
| Compression | Len. | 10.14 | 10.39 | 10.28 | 10.35 | 10.32 | **10.24** |
|  | Gap | 0.00% | 2.47% | 1.38% | 2.07% | 1.78% | **0.69%** |
| Cluster | Len. | 10.35 | 10.60 | 10.49 | 10.56 | 10.49 | **10.45** |
|  | Gap | 0.00% | 2.42% | 1.35% | 2.03% | 1.35% | **0.87%** |
| Expansion | Len. | 9.40 | 9.64 | 9.53 | 9.61 | 9.61 | **9.50** |
|  | Gap | 0.00% | 2.55% | 1.38% | 2.23% | 2.23% | **0.85%** |
| Rotation | Len. | 9.43 | 9.66 | 9.56 | 9.62 | 9.58 | **9.53** |
|  | Gap | 0.00% | 2.44% | 1.38% | 2.01% | 1.59% | **0.74%** |
| Avg. Inf. Time (s) |  | 30 | 0.22 | 0.01 | 0.13 | 0.01 | 0.02 |

Table 4: Results of cross-distribution generalization on CVRP100

| Distribution | Metric | HGS | AM | POMO | MDAM | DROP | EL-DRL |
|---|---|---|---|---|---|---|---|
| Explosion | Len. | 14.30 | 14.79 | 14.65 | 14.76 | 14.70 | **14.62** |
|  | Gap | 0.00% | 3.43% | 2.45% | 3.22% | 2.80% | **2.24%** |
| Compression | Len. | 14.82 | 15.36 | 15.21 | 15.28 | 15.32 | **15.18** |
|  | Gap | 0.00% | 3.64% | 2.63% | 3.10% | 3.37% | **2.43%** |
| Cluster | Len. | 15.44 | 15.99 | 15.83 | 15.91 | 15.90 | **15.79** |
|  | Gap | 0.00% | 3.56% | 2.53% | 3.04% | 2.98% | **2.27%** |
| Expansion | Len. | 13.70 | 14.18 | 14.02 | 14.17 | 14.19 | **13.99** |
|  | Gap | 0.00% | 3.50% | 2.34% | 3.43% | 3.58% | **2.12%** |
| Rotation | Len. | 13.97 | 14.46 | 14.30 | 14.40 | 14.39 | **14.28** |
|  | Gap | 0.00% | 3.51% | 2.36% | 3.08% | 3.01% | **2.22%** |
| Avg. Inf. Time (s) |  | 30 | 0.29 | 0.03 | 0.17 | 0.05 | 0.08 |

## 4.2 Cross-distribution Generalization on CVRP

Pertaining to CVRP, we present the averaged tour lengths and gaps to the best solutions (obtained from the HGS with a 30s runtime), along with the average time to inference on unseen instances of CVRP50 and CVRP100 from five distributions in Tables 3 and 4, respectively. Overall, the HGS solver performs the best regarding tour length, as this heuristic is highly specialized for CVRP. Notably, we observe greater advantages of our method against neural baselines on this harder problem than on TSPs. Our EL-DRL achieves superior results on unseen instances of various distributions, with significantly small gaps (0.69-0.87%) on CVRP50, and especially outperforms POMO by around 0.7% on Compression distribution. Table 4 shows that for neural baselines, generalizing to various distributions on CVRP100 is also challenging, with considerable gaps ranging from 2.36% to 3.64%. In contrast, our method consistently delivers high-quality solutions with shorter lengths and is less sensitive to distribution shifts, which attains gaps of close to 2% for cross-distribution generalization on CVRP100. This superiority verifies the effectiveness of our idea of leveraging the strength of handling different distributions through cooperation between sub-policies. Regarding efficiency, all neural baselines can solve an instance in less than one second. Owing to our design of lightweight ensemble architecture and parallel inference, EL-DRL does not incur much inference time than the counterparts using a single model while yielding better performance and even exhibiting faster inference than AM and MDAM.

## 4.3 Real World benchmarks

We continue to evaluate our EL-DRL by taking public benchmark datasets TSPLib [53] and CVRPLib [54] as the testbed, to prove our approach is also efficient in solving more realistic and varied distributions. TSPLib is a library of TSP instances from various sources and of various types, making it desirable yet challenging for assessing generalization. For TSPLib, results are collected from 25 instances which problem sizes ranging from 50 to 300. For CVRPLib, the dataset is XML100, which

Table 5: Results on Real-World Benchmarks

| Dataset | Metric | Opt. | AM | POMO | MDAM | HAC | DROP | AMDKD | EL-DRL |
|---------|--------|------|----|------|------|-----|------|-------|--------|
| **TSPLib** | Len. | 6.86 | 8.02 | 7.44 | 7.73 | 8.65 | 7.48 | 7.45 | **7.42** |
| | Gap | 0.00% | 10.53% | 5.16% | 7.82% | 16.75% | 5.79% | 5.30% | **4.85%** |
| **Avg. Time (s)** | | - | 0.48 | 0.47 | 0.36 | 0.48 | 0.35 | 0.53 | 0.95 |
| **CVRPLib** | Len. | 16.97 | 17.82 | 17.70 | 17.76 | - | 17.84 | 17.67 | **17.65** |
| | Gap | 0.00% | 6.05% | 4.51% | 4.90% | - | 5.25% | 4.32% | **4.12%** |
| **Avg. Time (s)** | | - | 0.29 | 0.03 | 0.15 | - | 0.05 | 0.03 | 0.08 |

contains 10,000 CVRP100 instances from heterogeneous distributions. We normalize the coordinates in each instance to a range of [0, 1] for all neural methods, which also include the neural baseline AMDKD [15].

The upper half of Table 5 illustrates the averaged results on TSPLib instances. These results demonstrate that EL-DRL generalizes well to realistic distributions and varied sizes, in a small gap (4.85%), and it is much lower than the gaps of neural heuristics (i.e., around 5%-17%). In particular, the advantage of our EL-DRL over MDAM implies learning similar ensembles may hold back the generalization performance. Compared to the poor results of HAC, which learns only from Gaussian distribution, we can conclude that training the models on various heterogeneous distributions could be more beneficial. In comparison to the most recent generalization-specialized AMDKD, our EL-DRL brings a reduction in the gap by 0.51%. The lower half of Table 5 shows the averaged lengths and gaps on (harder) CVRPLib instances. Our EL-DRL again outperforms the top two baselines, AMDKD and POMO, by margins of 0.20% and 0.40%, respectively. The results on the realistic benchmarks underscore the remarkable capability of our EL-DRL in solving miscellaneous instances that were not present during the training phase, highlighting its robustness in coping with distribution shifts.

## 4.4 Ablation Studies

Table 6: Ablation Studies on CVRP100

| Method | POMO | Multi-POMO | EL-DRL | w/o BRI | w/o $\mathcal{L}_T$ | w/o $\mathcal{L}_P$ |
|--------|------|------------|--------|---------|--------|--------|
| Avg. Length | 14.801 | 14.784 | **14.747** | 14.768 | 14.752 | 14.756 |
| Avg. Gap | 2.423% | 2.311% | **2.055%** | 2.201% | 2.090% | 2.118% |

We also carry out ablation studies to demonstrate the effectiveness of key designs in our EL-DRL, by generating 10,000 CVRP100 instances as an exemplary case and presenting the results in Table 6. To investigate the disadvantage of high similarity between ensemble policies, we deploy a single POMO and an ensemble of four POMOs (Multi-POMO), respectively, to compare with our EL-DRL. Additionally, we ablate Bootstrap with Random Initialization (BRI), the Theil index regularization ($\mathcal{L}_T$) and parameter dissimilarity regularization ($\mathcal{L}_P$), then report the averaged results for the 10,000 instances of five distributions on CVRP100. First, the single POMO model relies on its own learning ability, increasing the susceptibility to over-fitting and leading to poor generalization performance. However, simply stacking POMO to multiple copies (Multi-POMO) cannot overcome this limitation and only brings slight improvement compared to the single POMO, which is primarily attributed to the larger amount of model parameters and in accordance with the finding in [60]. By contrast, our EL-DRL attains a 0.37% reduction of the overall gaps compared to the single POMO by collaboratively solving instances on various distributions. Excluding the Bootstrap with random initialization, sub-policies cannot receive distinct loss signals and converge to similar ones, and thus the gap of the variant without BRI increases significantly to 2.201% compared to that of EL-DRL (2.055%). Moreover, removing two regularization terms for maximizing diversity results in redundant parameters for sub-policies and hinders their expressive ability, as evidenced by the drop in generalization performance shown in the last two columns of Table 6. In summary, the designs of our EL-DRL are largely orthogonal and can be effectively integrated to maintain ensembles of diverse sub-models, which are capable of capturing different aspects of various distributions.

# 5   Conclusion and Limitation

This paper presents an ensemble-based DRL approach for solving vehicle routing problems, which fosters the capability of existing neural heuristics in generalizing against distribution shifts. By enforcing distinct reinforcement loss signals on each sub-policy and integrating regularization to ensure diversity across sub-policies, our EL-DRL learns to leverage the strengths of individuals in solving different instances with various distributions. In our experiments, we demonstrate that our proposed method delivers high-quality solutions for both synthetic instances of various distributions and heterogeneous instances from real-world benchmarks, confirming the potential of the ensemble idea in contributing to more generalizable and robust neural heuristics.

While it is not the focus of this paper, our method might be less effective in handling large instances. A major reason is that the scalability of our EL-DRL highly depends on the sub-model, where POMO (the used sub-model) performs inferior on large ones. Moreover, since our EL-DRL follows the ensemble paradigm, the training needs more computational resources than the single one. In the future, we would like to, 1) foster the scalability of our method by exploring decomposition schemes such as the divide-and-conquer [61], so as to allow it to solve large instances more efficiently; 2) enhance the computational efficiency of our method by introducing sparsity in ensemble learning [62], so as to allow it to train more sub-models while reducing the training overhead; 3) enable the interpretability [63] and further strengthen the collaboration among the sub-models of our method; and 4) extend our method to non-routing combinatorial optimization problems.

## Acknowledgments and Disclosure of Funding

This research was supported by the National Natural Science Foundation of China under Grant 62102228, the Singapore Ministry of Education (MOE) Academic Research Fund (AcRF) Tier 1 grant, and the RIE2025 MTC IAF-PP funding (M23L4a0001).

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
