# OpenReview forum: "Ensemble-based Deep Reinforcement Learning for Vehicle Routing Problems under Distribution Shift"
_NeurIPS.cc/2023/Conference — NeurIPS 2023 poster_

### Official Review · Reviewer_eih2 · 2023-06-22

**Soundness:** 4 excellent
**Presentation:** 3 good
**Contribution:** 4 excellent
**Rating:** 7
**Confidence:** 4

**Summary:**

This paper suggests a novel method to mitigate the distributional shift of combinatorial optimization problems. The idea is to bootstrap and aggregate multiple diverse policies which are similar to a classical ensemble method called bagging. The idea is simple and effective having promising empirical performances. I clearly vote to accept this paper.


**Strengths:**

They targeted one of the significant research topics of combinatorial optimization. The idea is simple and straightforward. The empirical results seem to be effective.

**Weaknesses:**

One of the other major topics of the distributional shift of combinatorial optimization is scale shift (e.g., trained on $N=50$ and tested on $N=500). This paper did not discuss those topics.

**Questions:**

1.  I'm curious about the robustness of this algorithm on large-scale problems.

2.  There were several pieces of literature to mitigate distributional shift; for example:
- Sym-NCO (NeurIPS'22) [1]: leveraged symmetricity for better generalization
- ROCO (ICLR'23) [2]: robustness training of neural combinatorial optimization solver
- Meta-SAGE (ICML'23) [3]: leveraged meta-learning for better adaptation
In my humble opinion, this paper should include the recent rapid progress of related research more carefully.

3. Can this method be applied to non-routing problems?

[1] Kim, Minsu, Junyoung Park, and Jinkyoo Park. "Sym-NCO: Leveraging Symmetricity for Neural Combinatorial Optimization." Advances in Neural Information Processing Systems.

[2] Lu, Han, et al. "ROCO: A General Framework for Evaluating Robustness of Combinatorial Optimization Solvers on Graphs." The Eleventh International Conference on Learning Representations. 2023.

[3] Son, Jiwoo, et al. "Meta-SAGE: Scale Meta-Learning Scheduled Adaptation with Guided Exploration for Mitigating Scale Shift on Combinatorial Optimization." arXiv preprint arXiv:2306.02688 (2023).


**Limitations:**

They already stated their limitation regarding scalability.

---

> ### Author Rebuttal · Authors · 2023-08-09
>
> Thanks for reviewing our paper with positive feedback. We appreciate that the reviewer acknowledges the motivation and significance of our work. Our responses to the comments are as follows.
>
> Q1: One of the other major topics of the distributional shift of combinatorial optimization is scale shift (e.g., trained on $N=50$  and tested on $N=500$). This paper did not discuss those topics. I'm curious about the robustness of this algorithm on large-scale problems.
>
> A1: In this work, we focus on the topic of distribution shift. In our experiments on TSPLib (Table 5), we reused our trained $N=100$ models and tested on instances with problem sizes ranging from $N=50$ to $N=300$. Following your suggestion, we also reuse $N=50$ models and test TSPLib instances with problem sizes ranging from $N=50$ to $N=500$. The result is as below (We also include Sym-NCO):
> | Dataset  | Metric    | Opt.  | AM     | POMO  | MDAM   | HAC    | DROP  | AMDKD | SYM-NCO | EL-DRL |
> |----------|-----------|-------|--------|-------|--------|--------|-------|-------|---------|--------|
> | TSPLib   | Avg. Len. | 7.64  | 8.96   | 8.23  | 8.45   | 9.04   | 8.34  | 8.32  | 8.20    | 8.11   |
> | N~50-500 | AVG. Gap  | 0.00% | 17.28% | 7.72% | 10.60% | 18.32% | 9.16% | 8.90% | 7.33%   | 6.15%  |
>
> According to the experiment, our method surpasses other baselines by a larger margin including Sym-NCO regarding robustness on larger scale problems.
>
> Q2: There were several pieces of literature to mitigate distributional shift.
>
> A2: Thank you for these valuable works. For Sym-NCO, the original paper tested on TspLib instances with up to 250 nodes. We showed its results in the above table and our method outperforms this generalization-specific method. ROCO and Meta-SAGE are the most recent works released close to our submission date to NeurIPS 2023. For ROCO, the attacker only modifies the original by ‘choose an edge and half its value’, which does not create distribution shifts as in our scenarios. Meta-SAGE is ‘for improving the scalability of deep reinforcement learning models’ and has special components for larger-scale problems. Although it may not be fair to compare them directly, we are interested in their results on our cross-distribution instances after they share the source code. Also, we think our method can incorporate these scaling techniques when needed for large-scale problems. Since research on generalization for both distribution and scale is scarce, we agree that it is a good future research area. We will add the discussion on these works to our revision.
>
> Q3: Can this method be applied to non-routing problems?
>
> A3: Yes. As you summarized, our work is to propose a new framework for making decisions by multiple diverse policies for combinatorial problems. This idea is also applicable to other problems in which the testing distribution(pattern) is different from train data. For example, since our deep model backend is Attention Model [1, 2], we can reuse the neural net developed for TSP and apply it to solve 0-1 Knapsack Problem (KP) with minor changes like selected items (i.e. visited nodes) as well as items that no longer fit inside the knapsack are masked. Also,  the training reward should be changed to the total value of selected items.
>
> *We apologize for the character limit and plese refer to the  ‘author response’ for full reference list.*

---

> > ### Comment · Reviewer_eih2 · 2023-08-10
> >
> > Thank you for your timely response. I especially value the inclusion of the supplementary experimental results showcasing the promising potential of EL-DRL. With this in mind, I maintain a positive outlook on my assessment.

---

> > > ### Author Response · Authors · 2023-08-13
> > >
> > > We sincerely appreciate the reviewer for acknowledging our work and response.

---

### Official Review · Reviewer_WvQ5 · 2023-07-05

**Soundness:** 3 good
**Presentation:** 3 good
**Contribution:** 3 good
**Rating:** 5
**Confidence:** 4

**Summary:**

This paper investigated the classical vehicle routing problems and proposed an ensemble-based deep reinforcement learning algorithm. Moreover, two regularization terms (Theil index and L2-dist) were designed to improve the generalization ability of learned policies. The motivation is convincing and the proposed methodology is easy to understand.

**Strengths:**

1、The classical vehicle routing problem is important to explore and the proposed ensemble-based deep reinforcement learning algorithm has been proven to handle such a critical problem;
2、The motivation is convincing and the paper writing is easy to understand.

**Weaknesses:**

The main concern of mine mainly focuses on the limited discussion and comparison between the existing/previous works and this work:
(1): Since the authors break down the optimization problem into sub-policies learning, why not just use multi-agent reinforcement learning to solve the problem? I didn't quite understand the necessity of ensemble learning. What is the relationship between ensemble learning and multi-agent policy in RL?

(2): The EL-DRL is built on the top of REINFORCE algorithm that is very old. Can EL-DRL use better baselines like PPO or PPG? Or this setting is indispensable?

(3): Some existing work in RL proposes to use data augmentation (e.g., observation augmentation and intrinsic reward) to improve generalization ability, which is very efficient and cheap [1][2]. Can the authors compare EL-DRL with these approaches?

**Questions:**

Q1: Can the authors explain the relevance between their work and "MAXIMIZING ENSEMBLE DIVERSITY IN DEEP REINFORCEMENT LEARNING" published at ICLR2022?

Q2: Is there any comparison between EL-DRL and traditional algorithms like genetic algorithms?

Q3: More detailed literature review of DRL-based methods for VRP is needed.

[1] Reinforcement Learning with Augmented Data
[2] State Entropy Maximization with Random Encoders for Efficient Exploration

**Limitations:**

Please see the weakness and question section.

---

> ### Author Rebuttal · Authors · 2023-08-09
>
> We thank the reviewer for the comments and support. We also appreciate that the reviewer acknowledges the quality and contribution of the paper.
>
> Q1: Why not just use multi-agent reinforcement learning to solve the problem? What is the relationship between ensemble learning and multi-agent policy in RL?
>
> A1: We can treat our ensemble learning framework as a type of multi-agent system. In RL terms, each sub-policy is an agent and they collaborate to decide the next city to visit. Unlike some other multi-agent RL research, our ensemble mechanism lets our agents work on the same goal synchronously. In many multi-agent systems, agents work asynchronously and face different environments. Sometimes, they may compete with each other. But in our routing problem, the problem is a simple sequence-generation process, where all agents have the same goal: choose the optimal next city until the whole sequence is generated. For routing problems, a complex system that puts each agent in different environments and phases is unnecessary. It may waste resources to maintain the population and not ensure that they seek global rather than local optima. In summary, we aim to devise a simple and efficient collaborative mechanism for agents. Our experiments show that our mechanism outperforms other methods without extra multi-agent techniques.
>
>
> Q2: Can EL-DRL use better baselines like PPO or PPG?
>
> A2: Earlier research [4, 5] uses an extra critic network to predict the baseline in REINFORCE algorithm, also known as A2C but requires extra resources to build another neural network. Plus, the prediction can be unreliable when facing unseen instances. Recent research also uses PPO to train the routing model [3]. But even with more complex RL algorithms, the improvement over [2] that uses a naive REINFORCE algorithm is marginal. Therefore, using a more complex RL algorithm does not guarantee significant improvement. A better practice is to tailor learning algorithms for routing problems. The REINFORCE algorithm we use in this paper is not naive. As stated in Policy Optimization with Multiple Optima (POMO) [1], the agents generate and collect $N$ trajectories that start from $N$ different nodes. The POMO baseline is a shared baseline that averages the reward of $N$ collected trajectories, which reduces variance in policy gradients compared to the greedy-rollout baseline [2]. This baseline can be computed efficiently compared to other baselines. Therefore, using REINFORCE with POMO baseline has been a good practice in DRL-based methods for VRP [6, 7, 8], rather than others like A2C, PPO or PPG.
>
> Q3: Some existing work in RL proposes to use data augmentation. Can the authors compare EL-DRL with these approaches?
>
> A3: We agree that it is very efficient and cheap to improve generalization ability, but the augmentation for images of these works cannot be directly applied to VRPs. Actually, we have already used augmentation during training and inference. As we said above, our backend model POMO starts the trajectory from each node during training. During inference, POMO produces multiple greedy trajectories by rotating each input instance and starting from each node. The final solution is the best one among all sampled trajectories. We will add more details about the augmentations and refer readers to POMO's augmentation mechanism [1].
>
> Q4: Can the authors explain the relevance between their work and "MAXIMIZING ENSEMBLE DIVERSITY IN DEEP REINFORCEMENT LEARNING" published at ICLR2022?
>
> A4: Their work (MED) trains models on toy simulation environments and six Atari games with Q-learning algorithms. As we stated in our paper, directly using deep Q-learning to play games with limited actions does not suit our problem, since estimating Q-values for VRPs with hundreds of actions (i.e., the next node to visit) is hard, and these methods perform worse than policy gradient. In contrast, we propose a new Policy Gradient framework that solves routing problems effectively. Both MED and ours use regulation terms from previous works [9, 10] and show their effectiveness in experiments, but their work only shows that regulation terms can improve diversity in ensembles. MED is not for more practical and complex problems like ours. In the Ablation Studies (Table 6), using a single term does not improve much. Our method improves from the careful design of the learning framework with mostly orthogonal components, not from enumerating each regularization term and testing it as MDE did.
>
> Q5: Is there any comparison between EL-DRL and traditional algorithms like genetic algorithms?
>
> A5: Some works compare genetic algorithms (GA) with DRL-based methods directly. However, the naive GA is not specifically tailored for VRPs and its search efficiency drops sharply as the problem size grows. As shown in Table 3 of [11], GA performs worse even with much longer runtime than DRL-based methods. Thus, in the literature, the common practice is to compare with strong traditional algorithms that are customized for VRPs. For example, since optimal solutions for CVRP are hard to obtain in a reasonable time, we use a strong meta-heuristic: Hybrid Genetic Search (HGS) as the baseline in our experiments (Table 3, Table 4 and Table 5). As a traditional algorithm, HGS also uses ideas from GA, but with more adaptations for VRPs. In summary, most existing DRL-based algorithms cannot outperform the strongest traditional algorithms. The advantage of DRL-based algorithms is the inference speed which can produce solutions in seconds. Our work aims to narrow the gap to stronger baselines like HGS than other DRL-based methods.
>
> Q6: More detailed literature review of DRL-based methods for VRP is needed.
>
> A6: Thank you for your suggestion. We will provide more details on the literature review of DRL-based methods for VRP in our revision for clarity.
>
> *We apologize for the character limit and plese refer to the  ‘author response’ for full reference list.*

---

### Official Review · Reviewer_L9hy · 2023-07-06

**Soundness:** 3 good
**Presentation:** 3 good
**Contribution:** 2 fair
**Rating:** 5
**Confidence:** 4

**Summary:**

In this paper, an ensemble-based deep reinforcement learning algorithm is proposed for vehicle routing problems. The goal of the method introduced in this work is to improve handling of distribution shifts at test time. The approach, EL-DRL, combines multiple ideas: i) and ensemble of sub-policies, ii) distinct losses for each sub-policy, iii) explicit regularization, and iv) distinct random sub-policy weight initialization. The main contribution of the work is a demonstration that sub-policy diversity improves out-of-distribution generalization for ensemble DRL on the VRP. The EL-DRL method is evaluated on a range of TSP and CVRP instances sampled from multiple distributions. It is compared against a significant number of advanced neural baselines and demonstrates improved performance.

**Strengths:**

#### Originality

- The main contribution of this paper appears to be a demonstration that diversity-enhanced Deep RL [1] can be applied to VRPs. I think this paper can be useful to those working on solving VRPs with Deep RL.

#### Quality

- The experiments are conducted on a wide range of benchmarks and out-of-distribution generalization is evaluation on a large number of synthetic and real VRP distributions.
- The authors did a good job placing this work in the context of the relevant literature on deep reinforcement learning for VRPs.

#### Clarity

- The paper is written well and easy to follow.

#### Significance

- The proposed approach EL-DRL, appears to achieve state-of-the-art results on a large number of experiments. However, there are some concerns about the interpretation of the results, detailed below.

#### References

[1] Sheikh, Hassam, Mariano Phielipp, and Ladislau Boloni. "Maximizing ensemble diversity in deep reinforcement learning." In International Conference on Learning Representations. 2021.

**Weaknesses:**

#### Originality

- It is not clear to me that the significance of this work extends beyond the applied research area of Deep RL for VRPs.
- I would assess the overall theoretical and technical contributions to the machine learning community to be incremental.

#### Soundness

- I struggled to interpret the quantitative results with any amount of confidence. While EL-DRL appears to "win" against the other methods (lots of bold), it was not clear to me whether the improvements due to the diversity-enhancing mechanisms are significant or marginal.
- I would highly recommend using rliable (https://github.com/google-research/rliable) to calculate boostrapped confidence intervals for summary statistics across multiple (5+, ideally) random seeds and to show performance profiles. The Probability of Improvement metric $P( X > Y)$  implemented in `rliable` is a great way to compare these methods against each other.
- On the majority of the tables in the results, POMO already beats the other baselines, including other advanced strategies for dealing with out-of-distribution generalization. This makes it difficult to interpret comparisons between EL-DRL and the other advanced strategies, as the strength of POMO alone appears to already explain the majority of the performance gains.
- For many settings, EL-DRL only achieves a slight improvement over POMO. For example, in Table 5 Real-World TSPLib, POMO gets 5.16% gap and EL-DRL gets 4.79% (a length difference of only 7.44 vs. 7.39).
- Moreover, I believe a random baseline and the Multi-POMO baseline should be included in each Table. A random baseline can help place into context what the difference in gap percentage means. Likewise, Multi-POMO will help provide context, as it has a similar number of parameters to EL-DRL. In Table 6, the Multi-POMO approach does improve over POMO (2.311% vs. 2.423%).

#### A minor issue

- L43-44: The authors write, "Motivated by the recognition of the limited capability for a single model to learn from multiple distributions, [...], we propose..." citing [1]. This statement is incorrect and should be revised. It is fairly clear by now that large pre-trained models are certainly capable of learning from multiple distributions. In support of this, the conclusions made in [1] are in fact that "increasing pre-trained model size consistently improves performance on Waterbirds and MultiNLI. We advise practitioners to use larger pre-trained models when subgroup labels are unknown".

#### References

[1] Pham, Alan, Eunice Chan, Vikranth Srivatsa, Dhruba Ghosh, Yaoqing Yang, Yaodong Yu, Ruiqi Zhong, Joseph E. Gonzalez, and Jacob Steinhardt. "The effect of model size on worst-group generalization." arXiv preprint arXiv:2112.04094 (2021).


**Questions:**

- How sensitive is the model to hyperparameters---e.g., the learning rate, $\alpha_1$ and $\alpha_2$?
- How does EL-DRL fare against random and Multi-POMO baselines on the other benchmarks beyond CVRP100?

---

Update after rebuttal: The authors have made an effort to respond in a satisfactory way to my concerns and questions. Under the assumption that the new experimental results (especially demonstrating the statistical significance of the improvement over baselines) and promised revisions make it into the final version, I have increased my score.

**Limitations:**

The authors mention two key limitations: EL-DRL inherits limitations of the sub-model, and the increase in computation required to use ensembles of multiple models. They mention that POMO performs inferior on large instances, although I did not see where this is demonstrated in the experiments?

---

> ### Author Rebuttal · Authors · 2023-08-09
>
> We appreciate your insightful and helpful feedback on our paper. Please find below our detailed responses to the concerns/comments raised by you.
>
> Q1: Verify whether the improvements due to the diversity-enhancing mechanisms are significant or marginal.
>
> As suggested, we used Google rliable to compute bootstrapped confidence intervals with 5 runs on TspLib with random seeds. The figure in the PDF file of the ‘Author Rebuttal’  shows the Probability of Improvement metric $P(Y \gt EL-DRL)$, where Y represents the baselines (including Multi-POMO). Since we aim for smaller values of tour length (higher probability for $Y \gt EL-DRL$), this metric demonstrates the significance of our EL-DRL in delivering better solutions consistently.
>
> Q2: On the majority of the tables in the results, POMO already beats the other baselines, as the strength of POMO alone appears to already explain the majority of the performance gains.
>
> POMO is a state-of-the-art neural solver which can serve as a strong baseline.  There are many works on the generalization of VRPs in the literature that use POMO [1] as their basic neural solver, which adds additional modules on top of POMO [6, 8, 13] to facilitate their generalization. Such generalization study is a trend in the research community, with significant value and importance. For example, DROP [6] employs distributional robust optimization to improve the generalization of POMO. AMDKD [8] distills the knowledge to train a POMO model from various pre-trained teacher models. To verify if employing POMO is a guarantee for generalization performance, we compare our model with POMO, DROP and AMDKD in our paper. In Table 5, even though DROP and AMDKD are specially designed for different distributions and built on top of POMO, they do not perform better than POMO or EL-DRL. DROP and AMDKD may excel on certain distributions, but they fall behind the original POMO on some others. This suggests that designing a suitable learning framework for generalizable models is more important than only building on top of POMO. Baselines with POMO usually outperform those without it, but EL-DRL beats all baselines including POMO itself in our experiment. For the analysis of the improvement over POMO, please see the answer of Q3.
>
>
> Q3: For many settings, EL-DRL only achieves a slight improvement over POMO. For example, in Table 5 Real-World TSPLib, POMO gets 5.16% gap and EL-DRL gets 4.79% (a length difference of only 7.44 vs. 7.39).
>
> Firstly, the experiment on TspLib is more difficult than others since we use models trained on $N=100$ and test on $N = 50$ to $N = 250$. In other words, we need to generalize our method for both distributions and scales at the same time.  In our new experiment that trains the model on $N = 50$ and test on $N = 50$ to $N = 500$, EL-DRL surpasses other baselines by a larger margin:
> | Dataset  | Metric    | Opt.  | AM     | POMO  | MDAM   | HAC    | DROP  | AMDKD | EL-DRL |
> |----------|-----------|-------|--------|-------|--------|--------|-------|-------|--------|
> | TSPLib   | Avg. Len. | 7.64  | 8.96   | 8.23  | 8.45   | 9.04   | 8.34  | 8.32  | 8.11   |
> | N~50-500 | AVG. Gap  | 0.00% | 17.28% | 7.72% | 10.60% | 18.32% | 9.16% | 8.90% | 6.15%  |
>
> We can observe that EL-DRL reduces the gap of POMO by 1.57%.
>
> Secondly, the theory of ‘No Free Lunch’ states that it is very hard to achieve an optimal solution when the testing distribution changes a lot, and nearly impossible to get a very low gap value for models that are assumed to be trained and tested in i.i.d. data. The lower the gaps, the harder it is for new methods to improve. For example, when the gap is large (AM and HAC), newer methods (POMO) can reduce the absolute gap value more easily. But when the gap is close to 5%, it is harder for new methods to achieve a much smaller gap. This phenomenon is common in machine learning research. Hence, our method's improvement to POMO is not marginal.
>
>
> Q4: Moreover, I believe a random baseline and the Multi-POMO baseline should be included in each Table. Likewise, Multi-POMO will help provide context, as it has a similar number of parameters to EL-DRL.
>
> Thanks for your advice. We will follow your advice that add a random and the Multi-POMO baseline to each Table. It will be more helpful for understanding how our design outperforms simply employing multiple submodels.
>
>
> Q5: A minor issue L43-44: It is fairly clear by now that large pre-trained models are certainly capable of learning from multiple distributions.
>
> Thank you for pointing this out. We agree that enlarging a single model may improve its performance and we will revise these sentences accordingly.
> However, only enlarging a single model does not guarantee meaningful improvement. For independent researchers, using a large amount of computational resources to build a huge model is impractical in most cases. Our goal is to improve the generalization performance with similar magnitude of computational resources. Furthermore, many researchers including the creator of PyTorch posted tweets to support the idea that GPT-4 may be a kind of ensemble learning of different smaller models. In other words, most famous artificial general intelligence models may also use the idea of ensembling to enhance their generalization ability on different tasks instead of only enlarging a single model.
>
> *We apologize for the character limit and plese refer to the  ‘author response’ for full reference list.*

---

> > ### Comment · Reviewer_L9hy · 2023-08-13
> > **Thanks for the response**
> >
> > The authors have made an effort to respond in a satisfactory way to my concerns and questions. Under the assumption that the new experimental results (especially demonstrating the statistical significance of the improvement over baselines) and promised revisions make it into the final version, I have increased my score.

---

> > > ### Author Response · Authors · 2023-08-13
> > >
> > > Thanks for your acknowledgment and support. We will revise our paper according to your advice.

---

### Official Review · Reviewer_PAi5 · 2023-07-26

**Soundness:** 2 fair
**Presentation:** 3 good
**Contribution:** 2 fair
**Rating:** 5
**Confidence:** 4

**Summary:**

The paper proposes a deep reinforcement learning approach for vehicle routing problems under distribution shift that trains an ensemble of models. During solution construction, all ensemble models are evaluated at each decision step and the actions are selected based on the combined output probabilities. The ensemble is trained and evaluated on instance sets that contain instances sampled from different distributions. Ideally, the individual models have different strength and weaknesses (e.g., with some performing well on uniform instances, while others are better at Gaussian instances) which could result in an improved generalization performance. In order to improve diversity between the ensemble models, the authors propose 3 different components: 1) Bootstrap with random initialization: All models are initialized with different parameter values and during training each model only sees a subset of the training data. 2) Theil index regularization: Aims at increasing the inequality between sub-policies. 3)  Parameter dissimilarity regularization: Aims at increasing the difference between the model parameters. The results indicate that the proposed ensemble approach performs better than comparable approaches.

**Strengths:**

- The paper addresses an important problem (designing a neural combinatorial optimization approach that generalizes to instances from different distributions).
- The results show that the method outperforms other methods in the considered generalization setting.
- The authors perform some ablation experiments to evaluate the effectiveness of the proposed diversity-increasing components.

**Weaknesses:**

- Overall, I am not fully convinced that the concept of combining outputs of multiple models to create a single solution is ideal for the neural combinatorial optimization setting where we usually generate multiple solutions per instance. In the classical machine learning prediction setting, model outputs are combined because users are interested in a single prediction. In the considered neural combinatorial setting there is no direct need to combine model outputs, instead one solution could be generated per ensemble model.
- While the authors performed some ablation experiments, I would have wished for even more experiments that provide insights into the method:
     - Some experiments that directly evaluate how different the policies of the ensemble models are would provide interesting insights. For example, how often do the models disagree on the next actions (if argmax is used to select an action separately for each model).
    - What is the impact of the weights for the two regularization terms (which are set to 0.5 and 0.5 in the experiments)? A parameter sweep that finds the sweet spot between too much and too little diversity would have been nice.
    - What is the impact of the ensemble size on the performance?
- The description of the proposed regularization terms and their theoretical motivation is difficult to understand (or lacking). I do not understand the Theil index regularization and I am skeptical that the parameter dissimilarity regularization meaningfully increases the difference between models. The experimental results do not fully convince me that they work as intended because it is unclear if the reported improvements are statistically significant. The authors should expand the description of the regularization terms, provide a more mathematically-grounded motivation, and more experiments.
- The experimental setup is a bit unclear. More specifically, the generation of the training instances is not explained in detail and it is unclear from how many different distributions instances are sampled. Furthermore, it is unclear if all test instances are sampled from distribution not seen during training or if there is some overlap.
- The runtime of the proposed approach is roughly two times slower than that of the POMO model. This makes a direct comparison of the cost difficult. What happens if the number of generated solutions for POMO is increased (e.g., doubled)?

**Questions:**

How many solutions do you generate per test instance?

**Limitations:**

The authors briefly discuss limitations of their approach.

---

> ### Author Rebuttal · Authors · 2023-08-09
>
> Thanks for reviewing our paper with constructive and valuable feedback. Our detailed responses to the concerns/comments are as follows.
>
> Q1:  How many solutions do you generate per test instance? In the considered neural combinatorial setting there is no direct need to combine model outputs, instead one solution could be generated per ensemble model.
>
> A1: Yes, it is ideal for generating multiple solutions per instance. Our EL-DRL uses the augmentation mechanism of POMO [1] during the inference process. It produces multiple greedy trajectories by rotating each input instance to generate $8 \times$ instances with different angles but the same optimal solution. For each generated instance, it starts the trajectory from each of the $N$ nodes. Thus, it samples $8 \times N$ solutions in total for each input instance. For example, EL-DRL samples $8 \times 100 = 800$ trajectories (solutions) for an instance with 100 nodes. The final solution is the best one among all the sampled trajectories. We will add more details about the augmentations into the revised version of our paper, and refer readers to POMO's augmentation mechanism [1].
>
> Q2: How often do the models disagree on the next actions (argmax)?
>
> A2: Thanks for your question. This experiment will be helpful for examining the effectiveness of our design for diversifying.
>
> | Problem    | TSP100 | CVRP100 |
> |------------|--------|---------|
> | Multi-POMO | 10.29% | 14.83%  |
> | EL-DRL     | 24.57% | 32.95%  |
>
> In this table, we present the average probabilities of the models that disagree on the next actions. When simply assembling multiple POMOs, these sub-models tend to be similar and cannot produce diverse opinions. In contrast, our mechanisms for enhancing diversity successfully diversify the ensemble and enable sub-models to play their strength in different situations.
>
> Q3: What is the impact of the weights for the two regularization terms?
>
> A3: We swept the weights in grids (i.e. [0.05, 0,5, 1, 2]) for both terms. We find that large diversity (weights are 1 or 2) makes sub-policies favor diversity over higher reward, which hinders the overall ensemble performance. Small diversity (weights are small like 0.05) makes sub-policies produce similar decisions and loses the population benefit. Setting weights to 0.5 for the two regularization terms is a good balance between the objective quality and diversity. Fine-grained tuning for these hyperparameters may improve the results slightly, but it is not the focus of this paper.
>
>
> Q4: What is the impact of the ensemble size on the performance?
>
> A4: We presented an ablation study on the number of sub-policies in Section D (Figure 2) of the original Supplementary Material. The average gaps on CVRP100 with five distributions show that the average gap reduces from 2.048 to 1.960 as the number of sub-policies increases from 3 to 6. However, the gain is the biggest when this number increases from 3 to 4, and then adding more sub-policies only delivers a slight boost (only about 0.002-0.003 gap reduction). Therefore, to achieve a good trade-off between computational overhead and performance, we recommend using 4 sub-policies. However, in the future, we will investigate how to efficiently assemble more sub-policies.
>
> Q5: The authors should expand the description of the regularization terms, provide a more mathematically-grounded motivation, and more experiments.
>
> A5: We will provide more description for these terms. Theil index is a measure of inequality that is based on the concept of entropy in information theory. Entropy measures the amount of uncertainty or randomness in a system. The more distinction the distribution, the higher the Theil index. The Theil index can be interpreted as the average logarithmic distance between the incomes or wealth of individuals and the mean income or wealth of the society. In contrast to conventional approaches, we aim to maximize the ensemble diversity in EL-DRL, thus encouraging inequality within the system.
>
> For the statistical significance (rliable [14]), the figure in the PDF file of the ‘Author Rebuttal’ demonstrates the significance of our approach with regularization terms over simply using multiple sub-models (Multi-POMO).
>
> Q6:  Explain generation of the training instances and how test instances are sampled.
>
> A6: To ensure the hardness and variety of training distributions and to guarantee that the training instances are not seen during testing, we generate 7.2 million training instances from mixed distributions. Data generation is done by first sampling a uniform instance and then **non-repetitively applying three random mutation operators** of TSPGEN to the same instance. So a single training instance is of **mixed distributions** and it does not belong to any single distributions in the testing.
>
> During the inference, we evaluated all methods on five unseen distributions. We test baselines on 10,000 instances in total, with each of the five testing distributions containing 2,000 instances. As stated in the response to Q1, we generate $8 \times N$ solutions per test instance.
> In short, the training instances do not fall into any type of testing distribution. All test instances are sampled from distribution not seen during training.
>
> Q7: What happens if the number of generated solutions for POMO is increased (e.g., doubled)?
>
> A7: We modify POMO in this experiment to use sampling for generating multiple solutions:
>
> | Model   | POMO  | POMO x2 | POMO x4 | EL-DRL |
> |---------|-------|---------|---------|--------|
> | CVRP100 | 2.63% | 2.57%   | 2.55%   | 2.12%  |
>
> We report the average gaps over 5 testing distributions on CVRP100 in the table above. Doubling (POMO x2) or quadrupling (POMO x4) the generated solutions does not add enough diversity and thus it is hard to find diverse solutions with the same model that produces the same probabilities for actions.
>
> *We apologize for the character limit and plese refer to the  ‘author response’ for full reference list.*

---

> > ### Comment · Reviewer_PAi5 · 2023-08-16
> >
> > Thank you for your response!
> >
> > > In this table, we present the average probabilities of the models that disagree on the next actions.
> >
> > Can you explain the results in more detail? I am not sure if I understand them correctly.

---

> > > ### Author Response · Authors · 2023-08-16
> > >
> > > We apologize for not providing a more detailed explanation in our previous response due to the character limit. We will further explain the experiment and results in the following.
> > >
> > > Specifically, in the last layer of each sub-model, the Softmax computation is used to output the probabilities of selecting each city separately for each sub-model. For instance, if we have three cities and two sub-models, the sub-model 1 may output [0.2, 0.5, 0.3] and the sub-model 2 may output [0.6, 0.1, 0.3]. When the *argmax* function is used to select an action separately for each sub-model, the sub-model 1 votes to visit city 2 and the sub-model 2 votes to visit city 1.
> > >
> > > We followed your suggestion to use the *argmax* function to determine the action with the highest probability for each sub-model. In this way, each sub-model may output the same or different actions based on their maximum probabilities. Specifically, when all sub-models output the same action (i.e., this action has the maximum probability for all sub-models), we mark it as *agree*. When the sub-models output different actions (i.e., at least any two of them vote for different actions), we mark it as *disagree*. Since we aim to encourage sub-models to deliver diverse solutions, the output of our proposed ensemble model (EL-DRL) should present more *disagree* than simply assembling multiple POMOs (Multi-POMO).
> > >
> > > To empirically verify the above point, we evaluated the trained models for TSP100 and CVRP100 for inferring solutions, and counted the frequency that sub-models output the same actions (agree) and different actions (disagree) at each step. The ratio of *disagree* was calculated using the formula $\frac{disagree}{agree+disagree}​×100$ for all steps of  each instance. We collected and averaged all ratios of disagreement on all test instances and reported the average ratios for EL-DRL and Multi-POMO in the table.
> > >
> > > In the table, we observe that when simply assembling multiple POMOs, the sub-models of Multi-POMO tend to be similar and cannot produce diverse opinions (with smaller *disagree* frequency). In contrast, our ensemble model enhances the diversity with larger frequencies. I.e., 10.29% vs. 24.57% for TSP 100; and 14.83% vs. 32.95% for CVRP100. Such diversity helps our ensemble model deliver better generalization performance than Multi-POMO, as shown by the results of our main paper.
> > >
> > > We hope that our response has clarified your questions. Please do not hesitate to contact us if you have any further questions.

---

> > > > ### Author Response · Authors · 2023-08-20
> > > >
> > > > Dear Reviewer PAi5
> > > >
> > > > We sincerely appreciate your valuable feedback. Given the author-reviewer discussion is approaching the deadline, could you please kindly let us know if you still have any other concerns? so that we could have enough time to respond to them. Thanks for understanding.
> > > >
> > > > Authors

---

> > > > > ### Comment · Reviewer_PAi5 · 2023-08-21
> > > > >
> > > > > Thank you for the more detailed explanation. This indeed indicates that models learn diverse policies. I increase my rating to 5.

---

> > > > > > ### Author Response · Authors · 2023-08-21
> > > > > >
> > > > > > Many thanks for rasing the score. We greately apprecite your valuable comments, and will incorporate your suggestions in the final version.

---

### Author Rebuttal · Authors · 2023-08-10

We appreciate all reviewers for their thoughtful reviews and constructive comments. In this general response, we provide the reference list, as well as a figure in the PDF file to support our response. Our detailed point-to-point responses can be found below.

## STATISTICALLY SIGNIFICANT

We used Google rliable Library [14] to compute bootstrapped confidence intervals with 5 runs on TspLib with random seeds. The figure in the PDF file of the ‘Author Rebuttal’ shows the Probability of Improvement metric $P(Y \gt EL-DRL)$, where Y represents the baselines (including Multi-POMO). Since we aim for smaller values of tour length (higher probability for $Y \gt EL-DRL$), this metric demonstrates the significance of our EL-DRL in delivering better solutions consistently.

## REFERENCE

[1] Kwon, Yeong-Dae, et al. "Pomo: Policy optimization with multiple optima for reinforcement learning." Advances in Neural Information Processing Systems 33 (2020): 21188-21198.

[2] Kool, Wouter, Herke van Hoof, and Max Welling. "Attention, Learn to Solve Routing Problems!." International Conference on Learning Representations. 2018.

[3] Wan, Ching Pui, Tung Li, and Jason Min Wang. "RLOR: A Flexible Framework of Deep Reinforcement Learning for Operation Research." arXiv preprint arXiv:2303.13117 (2023).

[4] Nazari, Mohammadreza, et al. "Reinforcement learning for solving the vehicle routing problem." Advances in neural information processing systems 31 (2018).

[5] Bello, Irwan, et al. "Neural Combinatorial Optimization with Reinforcement Learning." International Conference on Learning Representations. 2017.

[6] Jiang, Yuan, et al. "Learning to solve routing problems via distributionally robust optimization." [7] Proceedings of the AAAI Conference on Artificial Intelligence. Vol. 36. No. 9. 2022.

[7] Hottung, André, Yeong-Dae Kwon, and Kevin Tierney. "Efficient Active Search for Combinatorial Optimization Problems." International Conference on Learning Representations.

[8] Bi, Jieyi, et al. "Learning generalizable models for vehicle routing problems via knowledge distillation." Advances in Neural Information Processing Systems 35 (2022): 31226-31238.

[9] J Johnston. H. theil. economics and information theory, 1969.

[10] Stephen Boyd, Neal Parikh, Eric Chu, Borja Peleato, Jonathan Eckstein, et al. Distributed optimization and statistical learning via the alternating direction method of multipliers. Foundations and Trends® in Machine learning, 3(1):1–122, 2011

[11] Zhang, Zixian, Geqi Qi, and Wei Guan. "Coordinated multi‐agent hierarchical deep reinforcement learning to solve multi‐trip vehicle routing problems with soft time windows." IET Intelligent Transport Systems (2023).

[12] Vidal, Thibaut. "Hybrid genetic search for the CVRP: Open-source implementation and SWAP* neighborhood." Computers & Operations Research 140 (2022): 105643.

[13] Hottung, André, Yeong-Dae Kwon, and Kevin Tierney. "Efficient Active Search for Combinatorial Optimization Problems." International Conference on Learning Representations, 2022.

[14] Agarwal, Rishabh, et al. "Deep reinforcement learning at the edge of the statistical precipice." Advances in neural information processing systems 34 (2021): 29304-29320.

---

### Decision · Program_Chairs · 2023-09-21

**Decision:**

Accept (poster)

**Comment:**

This submission received four reviews and all reviewer recommend acceptance, most with low score but consistent. Two reviewers increased their scores (to 5) after the rebuttal. This submission was then discussed in depth between the area and senior area chairs.

All reviewers find the paper well written and the experiments sufficient. Some extra experiments that study other distribution shift have been suggested and are included in the rebuttal. The main weaknesses that the reviewers point out are similarity to [40] and the specialised topic. However the authors convincingly answered to this in their rebuttal and are expected to revise their document accordingly.

In summary the paper presents a contribution to a hard and relevant problem and no reviewer recommends rejection. We recommend acceptance of the submission.